# **Evaluating the EPICC-Model for Regional Air Quality**

## Simulation: A Comparative Study with CAMx and CMAQ

2

1

- Mengjie Lou<sup>1</sup>, Qizhong Wu<sup>1</sup>, Wending Wang<sup>2,3,4</sup>, Huansheng Chen<sup>2,3,4</sup>, Kai Cao<sup>2</sup>,
- Xiaohan Fan<sup>5</sup>, Dingyue Liang<sup>5</sup>, FenFen Yu<sup>5</sup>, Jiating Zhang<sup>1</sup>, Wei Wang<sup>6</sup>, Zifa
- **Wang**<sup>2,3,4</sup>

- Institute of Earth System Science, Faculty of Geographical Science, Beijing Normal
- University, Beijing, 100875, China
- <sup>2</sup>State Key Laboratory of Atmospheric Environment and Extreme Meteorology,
- Institute of Atmospheric Physics, Chinese Academy of Sciences, Beijing, 100029,
- China
- <sup>3</sup>State Key Laboratory of Atmospheric Boundary Layer Physics and Atmospheric
- Chemistry, Institute of Atmospheric Physics, Chinese Academy of Sciences, Beijing,
- 100029, China
- 4University of Chinese Academy of Sciences, Beijing, 100049, China
- <sup>5</sup>3Clear Technology Co., Ltd., Beijing, 100029, China
- <sup>6</sup>China National Environmental Monitoring Centre, Beijing, 100012, China

- Correspondence: Qizhong Wu (wqizhong@bnu.edu.cn) and Wending Wang
- (wangwending@mail.iap.ac.cn)

2425

26 27

28

29 30

31

32 33

39

41

#### Abstract

This study presents a systematic evaluation of China's independently developed the EPICC-Model for regional PM<sub>2.5</sub> and MDA8 O<sub>3</sub> simulations against established international models, using unified WRF meteorological fields and a multi-source integrated emission inventory. Results highlight the strengths of the EPICC-Model in several aspects: it achieves relatively high spatial consistency for PM<sub>2.5</sub>, with an annual index of agreement (IOA) of 0.80, and accurately captures pollution patterns in heavily polluted North China. It also demonstrates improved performance in simulating summer O<sub>3</sub> peaks, reducing maximum biases by more than 20 µg m<sup>-3</sup>, primarily through enhanced heterogeneous HONO formation and nitrate photolysis pathways that elevate OH concentrations, and it incorporates the CB6r5 mechanism to better represent biogenic VOC oxidation. The model exhibits the highest hit rate (45.6%) for identifying moderate PM<sub>2.5</sub> and moderate O<sub>3</sub> pollution events and successfully reproduces persistent compound pollution episodes. However, all models share common limitations, including insufficient capability in reproducing heavy pollution episodes, systematic underestimation of SO<sub>4</sub><sup>2-</sup>, and uncertainties in SOA-related OC simulations. Future improvements should focus on refining secondary aerosol chemistry, emission inventories, and boundary layer representations. This study has not only demonstrated the performance of the EPICC-Model against international benchmarks but also

1

provides guidance for improving regional and global air quality models.

## 1 Introduction

47

49

53

57

63

69

73

77

88

90

With rapid urbanization and industrialization. China faces increasingly severe multi-pollutant air quality challenges. Among them, fine particulate matter (PM<sub>2.5</sub>) and ozone (O<sub>3</sub>) are key threats to public health and ecosystems, and their coordinated control has become a priority. As a typical secondary pollutant, PM<sub>2.5</sub> is influenced by primary emissions, secondary formation from SO<sub>2</sub>, NO<sub>x</sub>, and VOCs, regional transport, and meteorology (Zhang et al., 2012; Jing et al., 2020). Emission control policies have reduced population-weighted PM<sub>2.5</sub> exposure by ~48% from 2013 to 2020 (Xiao et al., 2022), yet severe winter haze episodes persist in regions like Beijing-Tianjin-Hebei. In contrast, O<sub>3</sub> formation shows nonlinear dependence on NO<sub>x</sub> and VOCs and is highly sensitive to radiation, temperature, humidity, and the boundary layer dynamics (Li et al., 2019b; Gao et al., 2022). Despite PM<sub>2.5</sub> reductions, O<sub>3</sub> levels have risen steadily, with warm-season maximum daily 8-hour average (MDA8)  $O_3$  increasing by  $1.2 \pm 1.3$ ppb yr<sup>-1</sup> in major urban clusters during 2013-2022 (Wang et al., 2024). The combined pollution of PM<sub>2.5</sub> and O<sub>3</sub> exhibits complex spatiotemporal evolution patterns and formation mechanisms (Lyu et al., 2025; Zhu et al., 2023), posing significant challenges for sustained air quality improvement in China.

To better understand the formation of complex air pollution and assess control strategies, Chemical Transport Models (CTMs) are widely used for regional air quality analysis and policy evaluation (Li et al., 2021; Gao and Zhou, 2024). Representative CTMs include the Community Multiscale Air Quality Modeling System (CMAQ) developed by the U.S. Environmental Protection Agency (EPA), the Comprehensive Air Quality Model with Extensions (CAMx) developed by ENVIRON Corporation, WRF-Chem for regional-scale simulations, and GEOS-Chem, which is widely applied for global-scale studies. However, current CTMs still exhibit significant uncertainties in simulating PM<sub>2.5</sub> and O<sub>3</sub> under specific pollution scenarios in China, primarily manifested as systematic biases in peak concentrations, spatiotemporal distribution errors, and inaccuracies in capturing seasonal variability (Bessagnet et al., 2016; Chen et al., 2019). Against this backdrop, China has independently developed the Emission and atmospheric Processes Integrated and Coupled Community Model (EPICC-Model) to enhance capabilities in simulating complex pollution processes. Supported by the National Natural Science Foundation of China (Major Program) and the Earth System Numerical Simulation Facility (EarthLab), the model was officially released on November 8, 2024 (EPICC-Model Working Group, 2025) (for detailed description, see Section 2.1).

As a newly released CTM, the EPICC-Model has undergone preliminary single-model performance evaluations (EPICC-Model Working Group, 2025; Wang et al., 2025), but systematic multi-model comparisons under nationwide, multi-season conditions remain lacking. Multi-model intercomparison not only helps reveal structural differences among models in pollutant transport, chemical reactions, and meteorological feedbacks, but also serves as a key approach to quantifying simulation uncertainties and improving model robustness and reliability (Carmichael et al., 2008). Although substantial efforts have been made in evaluating CTMs over China and East Asia, such as the MICS-Asia phase studies on PM<sub>2.5</sub> and O<sub>3</sub> (Chen et al., 2019; Gong and Liao, 2019; Itahashi et al., 2020; Li et al., 2019a), regional comparisons in the Pearl River Delta (Wu et al., 2012), and recent multi-model analyses over eastern China (Gao et al., 2024), most focus on earlier-generation or established models. Likewise, large-scale assessments within CMIP5/6 have consistently revealed systematic biases in

simulating PM<sub>2.5</sub> and components across China (Li et al., 2020; Ren et al., 2024). Given the structural innovations and representations in the EPICC-Model, it is both timely and necessary to conduct a systematic comparison with established CTMs over China, in order to comprehensively assess its strengths and weaknesses and provide a scientific basis for subsequent model improvement and application.

Therefore, this study implements a consistent model intercomparison framework to systematically compare the simulation performance of three models (EPICC-Model, CMAQ, and CAMx) for PM<sub>2.5</sub> and O<sub>3</sub> concentrations over China in 2021, based on a unified meteorological driving field and a multi-source integrated emission inventory. The objectives are to comprehensively assess the capabilities of the EPICC-Model, identify common issues and their potential causes across multiple models, and propose targeted improvements. The paper is structured as follows: Section 2 introduces the configuration schemes of the three models, emission data sources, the WRF-based meteorological forcing, and the evaluation of meteorological simulations. Section 3 presents the research results, including comparative analyses of the spatiotemporal distribution simulations of PM<sub>2.5</sub> and O<sub>3</sub>, PM<sub>2.5</sub> chemical composition analysis, assessments of the Air Quality Index (AQI) and pollution forecast accuracy, as well as comparisons of model performance in simulating persistent pollution events. Section 4 summarizes the key findings and discusses future directions for model improvements and prospects.

### 2 Data and methods

#### 2.1 Overview of simulation domain, period, and CTM configurations

This study employs a two-level nested grid configuration with the model domain centered at  $(35^{\circ}N, 105^{\circ}E)$ . The outer domain covers East Asia with a 45 km horizontal resolution  $(228 \times 165 \text{ grid cells})$ , while the inner domain focuses on China at a 15 km resolution  $(465 \times 300 \text{ grid cells})$ , starting from grid point (36, 39) within the outer domain. All analyses and results are derived from the inner domain simulations. The simulation period extends from 12:00 UTC on 14 December 2020 to 18:00 UTC on 31 December 2021, with the initial 17.25 days (until 18:00 UTC on 31 December 2020) dedicated to model spin-up.

For a systematic intercomparison of models, this study employs consistent meteorological driving fields generated by the Weather Research and Forecasting (WRF) model and a harmonized multi-source emission inventory to drive year-long simulations using the EPICC-Model, CAMx, and CMAQ. The EPICC-Model is a three-dimensional tropospheric CTM independently developed in China (EPICC-Model Working Group, 2025). Featuring highly modular architecture and parallel computing capabilities, it simulates key physical and chemical processes including emissions, transport, gas-phase and heterogeneous reactions, aerosol thermodynamics, and dry/wet deposition. CAMx (Anon, 2020) is a versatile photochemical grid model widely used for air quality scientific assessments and policy support. It follows the "one atmosphere" approach, accommodating pollution simulations from urban to regional scales (Emery et al., 2024). CMAQ (Anon, 2021) is a continuously evolving opensource modeling platform featuring an open architecture and multi-processor parallel computing capabilities. It efficiently simulates air pollution processes including O<sub>3</sub>, particulate matter, toxic pollutants, and acid deposition (US EPA, 2021).

137

138 139

140

141

142143

149

**Table 1.** Key configuration schemes of the EPICC-Model, CMAQ, and CAMx.

|                           | EPICC-Model                                                | CAMx                                      | CMAQ                       |
|---------------------------|------------------------------------------------------------|-------------------------------------------|----------------------------|
| Model version             | v1.0                                                       | v7.0                                      | v5.3.3                     |
| Vertical layers           | $20\sigma_z$ layers                                        | $14\sigma_p$ layers                       | $14\sigma_p$ layers        |
| Horizontal advection      | Walcek (Walcek and<br>Aleksic, 1998)                       | PPM (Colella and Woodward, 1984)          | PPM                        |
| Vertical advection        | Walcek                                                     | PPM                                       | PPM                        |
| Horizontal diffusion      | Multi-scale                                                | Multi-scale                               | Multi-scale                |
| Vertical diffusion        | YSU (Hong et al., 2006)                                    | YSU                                       | ACM2 (Pleim, 2007)         |
| Gas-phase                 | CB6r5 (Yarwood et al.,                                     | CB05 (Yarwood et                          | CB6r3 (Emery et al.,       |
| mechanisms                | 2020)                                                      | al., 2005)                                | 2015)                      |
| Aqueous-phase chemistry   | RADM (Walcek and<br>Taylor, 1986)                          | RADM                                      | AQCHEM                     |
| Aerosol processes         | ISORROPIA v2.2<br>(Fountoukis and Nenes,<br>2007)          | CF/ISORROPIA v1.7<br>(Nenes et al., 1998) | AE7/ISORROPIA<br>v2.2      |
| Secondary organic aerosol | Two-product model (Pandis et al., 1992; Odum et al., 1997) | SOAP (Strader, 1999)                      | AE7/VBS                    |
| Dry deposition            | ZHANG03 (Zhang et al., 2001, 2003)                         | ZHANG03                                   | M3DRY                      |
| Wet deposition            | Henry's law (William, 1803)                                | Henry's law                               | AQCHEM                     |
| Photolysis                | Streamlined TUV (Emery et al., 2010)                       | Streamlined TUV                           | Fast-J (Wild et al., 2000) |
| Boundary conditions       | MORZART                                                    | Default                                   | Default                    |

The key configuration parameters of the EPICC-Model, CAMx, and CMAQ are summarized in Table 1. The EPICC-Model employs a  $20\sigma_z$  layers coordinate system, demonstrating superior vertical resolution compared to the  $14\sigma_p$  layers coordinates used in other two models. Coupled with the YSU boundary layer scheme, this configuration enhances simulation accuracy for near-surface turbulent mixing and nocturnal stable layer structures. CMAQ utilizes the Asymmetric Convective Model, version 2 (ACM2) scheme, which exhibits stronger coupling with surface heat flux feedback mechanisms, particularly advantageous for simulating boundary layer evolution under environments with pronounced diurnal temperature gradients. CAMx retains the YSU scheme, maintaining an optimal balance between computational efficiency and precision (Jia and Zhang, 2020; Shi et al., 2021). In terms of chemical mechanisms, the EPICC-Model employs the CB6r5 gas-phase mechanism coupled with RADM aqueous-phase chemistry, allowing the representation of gas-aqueous reactions contributing to secondary organic aerosol (SOA) formation, particularly under highhumidity conditions where aqueous-phase reactions can influence SOA production (Yarwood et al., 2020). CMAQ uses the CB6r3 mechanism in combination with the AE7/VBS module to represent SOA formation, and from version 5.2 onward,

incorporates additional parameterizations such as Potential Combustion SOA (PCSOA) to compensate for combustion-related SVOC and IVOC emissions not captured in current inventories (Murphy et al., 2017). CAMx uses the CB05 gas-phase chemical mechanism coupled with the SOAP module to parameterize SOA formation, simulating the oxidation of both anthropogenic and biogenic VOCs and their partitioning into the aerosol phase. For photolysis rate calculations, both the EPICC-Model and CAMx employ the Streamlined TUV scheme to reduce computational cost, whereas CMAQ utilizes the Fast-J scheme, which offers more detailed representation of radiative shielding under high aerosol loading conditions (Barnard et al., 2004). Regarding boundary conditions, the EPICC-Model incorporates MOZART outputs as the initial and lateral background fields, while CMAQ and CAMx rely on their respective default settings. These differences in physical parameterizations and chemical mechanisms constitute a critical foundation for interpreting the divergent simulation results of PM<sub>2.5</sub> and MDA8 O<sub>3</sub> across the three models.

#### 2.2 WRF model configuration and meteorological simulation evaluation

Key configuration parameters of the WRF model used in this study are summarized in Table 2. The simulations were conducted using WRF version 3.9.1, with initial and boundary conditions derived from the National Centers for Environmental Prediction (NCEP) Final  $1^{\circ} \times 1^{\circ}$  reanalysis data (FNL, ds083.2), featuring a temporal resolution of 6 hours. To enhance the accuracy of the WRF simulations, four-dimensional data assimilation (FDDA) grid nudging was applied during the simulation process.

**Table 2.** Key configuration parameters of the WRF model.

| WRF v3.9.1                            |                                                      |  |  |  |  |
|---------------------------------------|------------------------------------------------------|--|--|--|--|
| Horizontal resolution                 | 45 km-15 km (one-way nested)                         |  |  |  |  |
| Number of sigma levels                | $30\sigma_p$ layers, with top layer at 50hPa         |  |  |  |  |
| Longwave Radiation                    | RRTMG (Iacono et al., 2008)                          |  |  |  |  |
| Shortwave Radiation                   | RRTMG                                                |  |  |  |  |
| Microphysics                          | Thompson (Thompson et al., 2008)                     |  |  |  |  |
| Land-surface Model                    | Unified Noah Land Surface Model (Tewari e al., 2004) |  |  |  |  |
| Advection                             | Monotonic transport                                  |  |  |  |  |
| Planetary boundary layer (PBL) scheme | YSU (Hong et al., 2006)                              |  |  |  |  |
| Cumulus option                        | Kain-Fritsch Scheme (Kain, 2004)                     |  |  |  |  |
| Nudging options                       | FDDA                                                 |  |  |  |  |

This study systematically evaluated the performance of the WRF model in simulating surface meteorological fields over mainland China for the year 2021, based on comprehensive observational data. The evaluation utilized daily observations from representative stations in 335 cities nationwide (excluding Jiayuguan and Wujiaqu due to data unavailability), covering key meteorological variables such as 2 m air temperature, 2 m relative humidity, 10 m wind speed, 10 m wind direction, precipitation, and surface pressure. To ensure seasonal representativeness, January, April, July, and

191

201202

212213

214

215

216

217

218219

220221

222223

224

225

226

October were selected to represent winter, spring, summer, and autumn, respectively, for comparative analysis (see Fig. S1- Fig. S24 and Supplementary Material for details, including the evaluation metrics).

The evaluation results indicate that the WRF model accurately captures the diurnal variations of key meteorological variables across most regions. For 2 m temperature simulations, the Pearson correlation coefficient (R) exceeds 0.7 in January and October, with the Root Mean Square Error (RMSE) below 5 °C and the Mean Bias (MB) constrained within ±2 °C in North, East, and South China. Simulations of 2 m relative humidity also performed well, with R values generally above 0.7 and RMSE below 15% in April and July; MB values in East and South China were less than  $\pm$  9%. However, relatively large humidity biases were observed in parts of Southwest and Northwest China, where MB in the southwest exceeded 15% and RMSE surpassed 17%. Previous studies suggest that simulations can be considered reliable when  $R \ge 0.6-0.7$ , RMSE for temperature ≤ 5 °C, and RMSE for humidity < 15% (Lou et al., 2025; Oyegbile et al., 2024). Therefore, the WRF model used in this study demonstrates high reliability in simulating temperature and humidity. Simulations of 10 m wind speed were generally stable, with R exceeding 0.7 and RMSE below 8 m/s in most regions. Nevertheless, systematic overestimations were observed in the Sichuan Basin, Xinjiang, and parts of Tibet. Although the wind speed RMSE exceeds typical values reported in some studies (4-6 m s<sup>-1</sup>), it remains within an acceptable range given the seasonal variability at the national scale (Xu et al., 2020; Yu et al., 2022b). In contrast, the performance for 10 m wind direction was relatively poor, with R values below 0.4 in most areas, except for moderate improvement in the Beijing-Tianjin-Hebei region, East China, Central China, and South China. This finding aligns with previous research noting WRF's general limitations in reproducing wind direction (Jiménez and Dudhia, 2013). For precipitation, correlation coefficients were generally above 0.4 nationwide. However, substantial errors were observed in parts of Southwest, Northwest, and South China, especially in regions with complex terrain, consistent with previous findings (Yu et al., 2022a). Surface pressure simulations showed strong stability, with R values exceeding 0.7 across the country. While parts of Southwest and Central China exhibited slight underestimation (MB < 20 hPa), surface pressure was slightly overestimated in Guizhou and Chongqing. Overall, the WRF model provides reliable meteorological forcing fields for CTMs.

#### 2.3 Emission inventory and observational data sources

The anthropogenic emissions data used in this study for China are derived from the Multi-resolution Emission Inventory for China (MEIC) developed by Tsinghua University, with a base year of 2019 (Geng et al., 2024). This inventory includes major source sectors such as transportation, industry, power generation, and residential combustion, and covers multiple pollutants including CO, SO<sub>2</sub>, NOx, VOCs, PM<sub>10</sub>, PM<sub>2.5</sub>, BC, and OC. Emissions of ammonia over China are obtained from the PKU-NH<sub>3</sub> inventory (Huang et al., 2012a; Kang et al., 2016), while emissions from biomass burning are taken from the "China Open Biomass Burning Emissions Inventory" with a base year of 2017 (Huang et al., 2012b; Song et al., 2009). The EDGAR v5.0 dataset

233234

235

236

237238

253254

255

256257

266267

268

(Crippa et al., 2020), with a base year of 2015 and a spatial resolution of  $0.1^{\circ} \times 0.1^{\circ}$ , was used to represent anthropogenic emissions outside China. Biogenic emissions are simulated online using the MEGAN v3.2 model (Guenther et al., 2012). It should be noted that although the base years of the emission inventories do not exactly match the simulation year of this study, the use of inventories from adjacent years remains a common practice in regional air quality modeling, especially in the absence of globally consistent, high-resolution, multi-sector emission datasets for the target year. Under the assumption of no substantial changes in regional climate and socioeconomic activities, such inventories can reasonably represent the emission patterns of the study period (Amnuaylojaroen et al., 2014; Huang et al., 2023; Wang et al., 2023).

All surface observational data used in this study were obtained from the China National Environmental Monitoring Center (CNEMC), including hourly PM<sub>2.5</sub> and O<sub>3</sub> concentrations from 1,644 national monitoring stations across China in 2021 (station distribution shown in Fig. S25). Based on these data, daily average PM<sub>2.5</sub> concentrations and MDA8 O<sub>3</sub> values were calculated for 337 cities. Additionally, aerosol chemical composition data were obtained from the same source for ten representative cities: Beijing (116.41°E, 40.04°N), Tianjin (117.21°E, 39.17°N), Zhengzhou (113.73°E, 34.77°N), Jinan (117.06°E, 36.66°N), Shanghai (121.53°E, 31.23°N), Nanjing (118.76°E, 32.07°N), Wuhan (114.37°E, 30.54°N), Fuzhou (119.31°E, 26.10°N), Chengdu (104.09°E, 30.64°N), and Chongqing (106.47°E, 29.62°N).

## 3 Results and discussion

## 3.1 Spatiotemporal distribution and statistical analysis of daily averages

Figure 1 presents the performance of the EPICC-Model, CAMx, and CMAQ in simulating annual PM<sub>2.5</sub> concentrations, evaluated against observations from 1,644 national monitoring stations across China. Based on the time series analysis (Figure 1a), the year can be divided into three distinct phases. Phase I (January to mid-May): During this period, both the EPICC-Model and CAMx significantly underestimated PM2.5 concentrations and failed to capture the three major pollution episodes at the beginning of the year, as indicated by the circles in the figure (with biases ranging from 25 to 52 μg m<sup>-3</sup>). In contrast, CMAO showed improved performance in capturing both the timing and intensity of pollution peaks. This was primarily attributed to its substantially higher simulated organic carbon (OC) levels relative to the other models and observations, which produced a compensatory effect on total PM<sub>2.5</sub> (Figure 6). Additionally, CMAQ exhibited intermittent overestimations during certain periods, which may be related to the ACM2 vertical diffusion scheme it adopts, where the minimum turbulent diffusion coefficient (Kz min) under stable boundary layer conditions is set too low (default nighttime Kz min = 0.01 m<sup>2</sup> s<sup>-1</sup>) (Kim et al., 2024). The contrasting pattern within the rectangular box was attributed to a dust event. Phase II (mid-May to mid-October): During this period, PM<sub>2.5</sub> concentrations were generally low and exhibited limited variability. This was primarily attributed to elevated planetary boundary layer heights, enhanced turbulent mixing, and frequent precipitation events during the summer, which collectively facilitated the dilution and wet deposition of near-surface particulate matter.

CAMx performed best in this phase. In contrast, both the EPICC-Model and CMAQ tended to overpredict pollutant levels, with CMAQ exhibiting a notably stronger positive bias. Phase III (mid-October to December): During the heating season, observed PM<sub>2.5</sub> levels increased significantly, accompanied by greater spatiotemporal variability. CAMx maintained relatively good simulation accuracy during this period, with smaller biases. However, the overestimation by CMAQ became more pronounced, which might result from intrinsic biases in its chemical and physical parameterizations, rather than from differences in emissions or meteorological inputs. The performance of the EPICC-Model was intermediate between the two.

**Figure 1.** Comparison of simulated and observed daily PM<sub>2.5</sub> concentrations across China in 2021. (a) shows the time series of daily PM<sub>2.5</sub>, with the black line representing observations from 1,644 national air quality monitoring stations and colored lines indicating different CTMs. (b) presents scatter plots of modeled versus observed PM<sub>2.5</sub>. All model outputs exclude dust aerosol contributions to isolate secondary particulate formation processes.

Statistical validation analysis (Figure 1b) further corroborates the above findings. Although CMAQ exhibits the highest correlation coefficient (R=0.85), most of its data points lie above the 1:1 reference line, indicating a systematic overestimation. CAMx demonstrates the best overall statistical performance, with a Fraction of Predictions Within a Factor of Two (FAC2) value of 0.94 and a Normalized Mean Bias (NMB) of merely -2.86%. It is important to note that CAMx struggles to capture severe pollution episodes, markedly underestimating periods of high concentrations, which constrains its usefulness for evaluating pollution processes. In contrast, the EPICC-Model shows greater adaptability. Its simulations during spring and summer are more consistent with CAMx, reflecting its accuracy under lower background concentration conditions. During the autumn and winter pollution seasons, its simulation trends align more closely with CMAQ, capturing pollution accumulation processes more effectively. This ability to respond to seasonal variations allows the EPICC-Model to strike a better balance between accurately simulating extreme pollution events and sustaining strong overall annual performance.

303

308

317

320

326327

328

Figure 2 presents the performance of the EPICC-Model, CAMx, and CMAQ in simulating MDA8 O<sub>3</sub> concentrations, evaluated against observations from 1,644 national monitoring stations across China. Observational data reveal a typical photochemically-driven seasonal pattern in O<sub>3</sub> concentrations: summer months exhibit peak levels due to enhanced VOC-NO<sub>x</sub> chain reactions under intense solar radiation and elevated temperatures (Seinfeld and Pandis, 2016; Sillman, 1999), while winter and spring maintain background concentrations resulting from reduced reaction activity associated with decreased O<sub>3</sub> photolysis rates  $(J_{O_2})$  and lower OH radical concentrations (Wang et al., 2019). The time series analysis (Figure 2a) shows that all three models reasonably reproduced the seasonal variation of O<sub>3</sub> concentrations, yet notable differences emerged across concentration levels. The EPICC-Model performed particularly well during high-concentration episodes, effectively capturing the peak levels of O<sub>3</sub> pollution. In contrast, both CAMx and CMAQ, while capable of reproducing high-concentration trends, systematically underestimated peak magnitudes with maximum negative biases reaching -20 μg m<sup>-3</sup>. This discrepancy mainly stems from differences in the treatment of photolysis and chemical mechanisms among the models. The EPICC-Model incorporated heterogeneous HONO formation and nitrate photolysis pathways that produce HONO, significantly enhanced the initial concentration of OH radicals. This accelerated the oxidation of VOCs and promoted rapid O<sub>3</sub> formation, thereby improving model performance during high-concentration periods (EPICC-Model Working Group, 2025; Wang et al., 2025; Zhang et al., 2022). In addition, the EPICC-Model employed the CB6r5 chemical mechanism, which offered more comprehensive representation of BVOC oxidation (especially isoprene) compared to CB6r3 (used in CMAO) and CB05 (used in CAMx), thereby increasing O<sub>3</sub> formation potential (Yarwood et al., 2020). Under low concentration background conditions, CMAQ captured the general trend reasonably well but exhibited a systematic positive bias, which may be attributed to its use of the ACM2 vertical mixing scheme that tends to produce an overly deep boundary layer and enhanced NO2 photolysis radical production (Hu et al., 2010). The EPICC-Model showed the smallest absolute bias but with a slightly weaker trend correlation.

Figure 2. Comparison of simulated and observed daily MDA8 O<sub>3</sub> concentrations across China in 2021.

Statistical analysis indicates that CAMx exhibited the highest correlation coefficient (R=0.94) with well-controlled absolute bias despite its systematic underestimation. CMAQ showed a slight overestimation throughout the year but maintains a strong temporal correlation (R=0.93), demonstrating good capability in capturing temporal variations. Notably, the EPICC-Model not only accurately captured peak concentrations but also achieved the lowest annual NMB, reflecting a more balanced and stable performance overall. The scatter plot (Figure 2b) further corroborates these findings: all three models show excellent FAC2 values with well-clustered data points. CMAQ data points predominantly lie above the 1:1 reference line, indicating a tendency to overestimate; CAMx points mostly fall below the line, indicating underestimation; whereas the EPICC-Model achieves a better balance between overestimation and underestimation, delivered the best overall simulation performance.

Figure 3 compares the simulated spatial distributions of PM<sub>2.5</sub> for the four seasons of 2021 produced by the EPICC-Model, CAMx, and CMAQ, revealing notable differences in regional simulation consistency and bias characteristics among the three models. As shown in 错误!未找到引用源。, the Index of Agreement (IOA) values vary across models and seasons, exhibiting clear seasonal patterns that quantitatively support the spatial distribution differences.

**Figure 3.** Seasonal spatial distributions of simulated PM<sub>2.5</sub> concentrations, with spring (March-May), summer (June-August), autumn (September-November), and winter (December-February).

355

356

357

358

359360

361

362363

364

365366

367

368

369

370371

372

373374

375

376377

382

385

392393

**Table 3.** IOA for the spatial distribution of PM<sub>2.5</sub> simulated by different models.

|             | Spring | Summer | Autumn | Winter | Year |
|-------------|--------|--------|--------|--------|------|
| EPICC-Model | 0.63   | 0.62   | 0.60   | 0.81   | 0.80 |
| CAMx        | 0.60   | 0.64   | 0.68   | 0.76   | 0.78 |
| CMAQ        | 0.72   | 0.58   | 0.54   | 0.73   | 0.77 |

In spring, under the influence of northwesterly winds, dust transport combined with local industrial emissions led to high PM<sub>2.5</sub> concentrations mainly distributed over North China, Northwest arid basins, and the Sichuan-Yunnan region. All three models effectively reproduced the pollution pattern, with IOA values exceeding 0.60, and CMAQ achieved the highest IOA of 0.72. Since dust concentrations were excluded from the particulate matter assessment in this study, all three models consistently underestimated PM<sub>2.5</sub> levels in the arid Northwest and southwestern plateau regions. In summer, influenced by the southeastern monsoon and intensive precipitation, PM<sub>2.5</sub> remain generally low across China with reduced regional variability. The three models exhibited similar spatial consistency nationwide, with IOA values ranging between 0.58 and 0.64. However, in the high-humidity coastal regions of South China, the EPICC-Model and CMAQ demonstrated slight advantages over CAMx in simulation accuracy. In autumn, the nationwide PM<sub>2.5</sub> concentrations rebounded due to an increase in stagnant weather conditions and a reduced boundary layer height. All three models showed varying degrees of overestimation in North China and the Sichuan Basin, likely related to ACM2/YSU schemes underestimating mixing layer height and weakening vertical diffusion at night (Jia et al., 2023). During this period, CAMx achieved the highest IOA value of 0.68, the EPICC-Model scores 0.60, while CMAQ dropped to its annual minimum of 0.54. In winter, PM<sub>2.5</sub> concentrations reached their annual peak driven by low temperatures, temperature inversions, subsidence, and heating emissions. All three models successfully reproduced the pollution patterns in polluted regions like North and Central China. The EPICC-Model demonstrated exceptional performance with an IOA of 0.81, while CMAQ and CAMx maintained strong consistency at 0.73 and 0.76, respectively. However, all models unrealistically simulate higher PM<sub>2.5</sub> concentrations in the Sichuan Basin compared to Henan Province. This discrepancy may have resulted from differences in emission estimates between the two regions: anthropogenic emissions in the Sichuan Basin are likely overestimated in the models, whereas frequent agricultural activities in Henan may lead to underestimation of NH<sub>3</sub> emissions, thereby suppressing nitrate aerosol formation. Previous studies have demonstrated that spatial errors in NH<sub>3</sub> emissions significantly impact nitrate-driven heavy pollution events (Kang et al., 2016; Kong et al., 2019; Liu et al., 2021).

Overall, all three models reliably simulate PM<sub>2.5</sub> spatial distributions, with annual mean IOA values above 0.77. the EPICC-Model performs best (IOA=0.80), showing highest consistency in winter and heavily polluted North China. CAMx is stable (IOA=0.78) and suitable for multi-seasonal averages but underestimates high-humidity regions and responds weakly to severe pollution. CMAQ shows slightly lower consistency (IOA=0.77), performs better in spring and winter, declines in summer and autumn, and exhibits a general positive bias, particularly in winter.

Figure 4 shows the seasonal spatial distribution of MDA8 O<sub>3</sub> in 2021 from the

three models. Given the strong seasonal variability of O<sub>3</sub>, model performance was quantitatively evaluated using the IOA metric, with detailed results provided in Table 4.

**Figure 4.** Seasonal spatial distributions of simulated MDA8 O<sub>3</sub> concentrations, with spring (March-May), summer (June-August), autumn (September-November), and winter (December-February).

Table 4. IOA for the spatial distribution of MDA8 O<sub>3</sub> simulated by different models.

|             | Spring | Summer | Autumn | Winter | Year |
|-------------|--------|--------|--------|--------|------|
| EPICC-Model | 0.56   | 0.79   | 0.74   | 0.77   | 0.85 |
| CAMx        | 0.53   | 0.85   | 0.70   | 0.76   | 0.87 |
| CMAQ        | 0.33   | 0.85   | 0.78   | 0.75   | 0.84 |

In spring,  $O_3$  pollution exhibits a multi-centered and scattered distribution pattern. The EPICC-Model reproduced this pattern well (IOA = 0.56) and outperforms CAMx and CMAQ, particularly in the high-concentration areas like Yunnan. CAMx generally underestimated  $O_3$  nationwide, likely due to limitations of the CB05 mechanism and its coarse  $14\sigma_p$  vertical resolution, which reduces mixing between near-surface precursors and O3. (Ren et al., 2022; Tang et al., 2011). CMAQ, in contrast, produced an unrealistic high- $O_3$  belt over the Yangtze River plain with a much lower IOA of 0.33. In summer, a high  $O_3$  concentration belt forms over the mid-latitudes (30° N ~40° N), and all three models successfully reproduced the enhanced  $O_3$  levels with the highest spatial consistency observed throughout the year. The summer IOA values for the EPICC-Model, CAMx, and CMAQ reach 0.79, 0.87, and 0.85, respectively. Given the use of identical meteorological fields, regional  $O_3$  levels are primarily controlled by photochemical production rather than constrained by boundary inputs (Li et al., 2019a).

In autumn, national O<sub>3</sub> declines as solar radiation weakens, with pollution centers shifting to eastern and southeastern coasts. All three models performed similarly (IOA 0.70–0.74). CMAQ performed best in the Pearl River Delta, accurately reproducing localized high concentrations, as supported by previous studies (Jiang et al., 2010; Wang et al., 2010). The EPICC-Model ranks second with slight overestimation in northern Jiangxi, and CAMx continues to underestimate O<sub>3</sub>. In winter, due to extremely weak solar radiation, reduced stratosphere-troposphere exchange, high NO<sub>x</sub> emissions, and limited boundary layer height, nationwide O<sub>3</sub> concentrations drop to their annual minimum and shift southward. All models perform steadily (IOA 0.75–0.77). The EPICC-Model captured accumulation over the coastal areas of southern China but slightly overestimates, CAMx shifted high concentrations southwest, and CMAQ overestimated in northern regions while missing the southward O<sub>3</sub> shift.

Comprehensive evaluation indicates that the EPICC-Model shows stable seasonal MDA8 O<sub>3</sub> performance with an annual IOA of 0.85, capturing O<sub>3</sub> distribution across China. CAMx has slightly higher consistency (IOA 0.87) but systematically underestimates in spring, autumn, and winter and misses localized peaks. CMAQ performs well in summer and autumn, especially over the Pearl River Delta, but IOA drops to 0.33 in spring and overestimates in winter, indicating a need to improve boundary layer and dry deposition schemes.

To comprehensively evaluate model performance, this study uses Taylor diagrams to compare the EPICC-Model, CAMx, and CMAQ simulations with observations on both annual and seasonal scales. The annual results (Figure 6a) show that all three models perform satisfactorily in simulating PM<sub>2.5</sub> and MDA8 O<sub>3</sub>, with particularly strong performance for MDA8 O<sub>3</sub>. Correlation coefficients exceed 0.75 for all models, while RMSE and STD remain relatively low, indicating higher stability and accuracy in MDA8 O<sub>3</sub> simulations compared to PM<sub>2.5</sub>.

**Figure 5.** Taylor diagrams of PM<sub>2.5</sub> and MDA8 O<sub>3</sub> for the three models on annual and seasonal scales. The angle represents R, the dashed arcs indicate RMSE, and the x and y axes represent STD. The "REF" point denotes the reference standard derived from observations.

Seasonally, PM<sub>2.5</sub> simulation accuracy exhibits marked variations: during winter and spring high-concentration periods, all three models show comparable performance,

with CMAQ demonstrating greater sensitivity to pollution peaks (STD $\approx$ 1, closest to observations). Model performance declines significantly under summer's low-background conditions (R<0.55, elevated RMSE/STD), indicating enhanced error sensitivity. Autumn sees performance recovery, with CAMx achieving R=0.97. In contrast, MDA8 O<sub>3</sub> simulations show more robust seasonal consistency: while spring remains the weakest season (R>0.7), the EPICC-Model and CMAQ maintain low errors. The EPICC-Model excels during summer O<sub>3</sub> peaks (R $\approx$ 0.95, lowest RMSE/STD), CMAQ leads in autumn, and all models converge in winter (R=0.8-0.9), effectively capturing O<sub>3</sub> spatiotemporal patterns.

Overall, the EPICC-Model demonstrates strong robustness in seasonal  $O_3$  simulations, maintaining consistently low standard deviations and showing clear advantages during the summer pollution peak. In contrast, CAMx and CMAQ exhibit relative strengths in  $PM_{2.5}$  simulations: CAMx performs particularly well under the complex meteorological conditions of autumn, while CMAQ better captures pollutant accumulation processes during the stable boundary layer conditions in winter, reflecting their respective adaptability.

#### 3.2 Evaluation of PM<sub>2.5</sub> chemical composition simulations

Based on the spatiotemporal variations of PM<sub>2.5</sub> total concentrations simulated by each model, the performance of the EPICC-Model, CAMx, and CMAQ in simulating major chemical components was further evaluated. Figure 6 presents the average relative contributions and absolute concentrations of key PM<sub>2.5</sub> components in ten representative cities across urban clusters, including the Beijing-Tianjin-Hebei, Yangtze River Delta, Chengdu-Chongqing, and the Middle Reaches of the Yangtze River regions. Overall, all three models reasonably reproduced the observed chemical composition structure characterized by a predominance of nitrate (NO<sub>3</sub><sup>-</sup>), which is consistent with previous findings (Huang et al., 2021). Observations indicated that NO<sub>3</sub><sup>-</sup> accounted for approximately 31.2% to 42.4% of PM<sub>2.5</sub> mass in most cities. This dominant feature was reproduced by all models, although the simulated concentrations tended to be overestimated to varying degrees.

Meanwhile, all three models consistently underestimated sulfate (SO<sub>4</sub><sup>2-</sup>), a bias consistent with existing research findings (Shao et al., 2019), which may stem from underestimating SO<sub>2</sub> oxidation rates or uncertainties in intra-cloud aqueous-phase chemical mechanisms. The insufficient formation of SO<sub>4</sub><sup>2-</sup> not only limits the production of ammonium sulphate [(NH<sub>4</sub>)<sub>2</sub>SO<sub>4</sub>] but also reduces the consumption potential of NH<sub>3</sub> (Gao et al., 2018), thereby suppressing the simulated concentration of ammonium (NH<sub>4</sub><sup>+</sup>).

For OC, although certain deviations are observed in individual cities, the overall simulation performance of the three models remains within an acceptable range (Miao et al., 2020). Among them, the EPICC-Model and CAMx produced results that are closer to observations, while CMAQ tends to systematically overestimate OC concentrations. This systematic bias may stem from the introduction of Potential Combustion SOA (PCSOA) species starting from CMAQv5.2. PCSOA serves as a surrogate secondary organic aerosol species to compensate for the lack of combustion-

related SVOC and IVOC emissions in current inventories. However, the model estimates PCSOA concentrations through parameterized precursor emissions and simplified processes including OH oxidation and gas-particle partitioning. The associated uncertainties may result in regional and temporal discrepancies, failing to reflect the real-world variations in PCSOA and consequently leading to simulation biases (Murphy et al., 2017; Pennington et al., 2021). Overall, all three models share substantial uncertainties in SOA-related mechanisms, representing a common limitation.

Black carbon (BC) is the least abundant component of PM<sub>2.5</sub>, with observed concentrations ranging from 0.7 to 2.1 µg m<sup>-3</sup>. All models exhibit generally consistent performance in simulating its relative contribution, with slight overestimations observed in Beijing, Tianjin, and Zhengzhou. These biases are primarily attributed to uncertainties in BC emission inventories, particularly from biomass burning and motor vehicle sources (Hong et al., 2017). Due to its low chemical reactivity in the atmosphere and limited removal through secondary processes, BC simulations are highly sensitive to emission inputs.

**Figure 6.** Comparison of simulated and observed aerosol components in representative Chinese cities. ASO<sub>4</sub>, ANO<sub>3</sub>, ANH<sub>4</sub>, OC, and BC denote sulfate, nitrate, ammonium, organic carbon, and black carbon in PM<sub>2.5</sub>, respectively. The locations of monitoring sites are as follows: BJ (Beijing; 116.41° E, 40.04° N), TJ (Tianjin; 117.21° E, 39.17° N), ZZ (Zhengzhou; 113.73° E, 34.77° N), JN (Jinan; 117.06° E, 36.66° N), SH (Shanghai; 121.53° E, 31.23° N), NJ (Nanjing; 118.76° E, 32.07° N), WH (Wuhan; 114.37° E, 30.54° N), FZ (Fuzhou; 119.31° E, 26.10° N), CD (Chengdu; 104.09° E, 30.64° N), and CQ (Chongqing; 106.47° E, 29.62° N).

As shown in the stacked bar charts of PM<sub>2.5</sub> absolute concentrations in Figure 6e-7h, Tianjin exhibits the highest levels among the selected cities, followed by Wuhan, while Fuzhou shows the lowest. This spatial pattern is closely related to regional emission intensity and meteorological conditions. The three models reasonably reproduce these concentrations in most cities, particularly in Nanjing. However, systematic biases exist in certain locations. Simulated concentrations in Tianjin are substantially lower than observations, primarily due to underestimation of OC

associated with uncertainties in SOA formation mechanisms. In contrast, concentrations in Chengdu and Chongqing are consistently overestimated across all models, indicating an amplified model response to pollution episodes in central China. Overall, while the models capture the major chemical composition features, their ability to accurately reproduce absolute concentrations and regional variability remains limited, highlighting the need for refined emission inventories and improved representation of meteorological processes.

**Figure 7.** Performance metrics of aerosol component simulations across models in representative Chinese cities.

Figure 7 presents boxplots of statistical evaluation metrics (R, RMSE, and NMB) based on ten representative cities to quantify the simulation accuracy and systematic biases of each model. Regarding the overall distribution of R values, a clear "stepwise" difference is observed among the three models. CMAQ generally exhibits higher R values across PM<sub>2.5</sub> components than the EPICC-Model and CAMx, indicating stronger capability in reproducing spatiotemporal variability. Among individual components, NO<sub>3</sub><sup>-</sup> shows the highest correlation (median R > 0.6) with no outliers, suggesting robust representation. NH<sub>4</sub><sup>+</sup> and OC display slightly lower R values, reflecting moderate uncertainty but acceptable overall performance. SO<sub>4</sub><sup>2-</sup> shows dispersed correlations and some low R values, indicating substantial simulation errors, likely due to insufficient representation of regional secondary aerosol formation. BC exhibits the most compact R distribution, with low inter-city variability, reflecting stable and consistent simulation by all models. The RMSE analysis shows larger errors for NO<sub>3</sub><sup>-</sup> and OC, with some

cities reaching 12 µg m<sup>-3</sup> and median values above 6 µg m<sup>-3</sup>, reflecting sensitivity to meteorology, secondary processes, and SOA-related uncertainties. The EPICC-Model performs well in this aspect. SO<sub>4</sub><sup>2-</sup> and NH4<sup>+</sup> exhibit moderate RMSE, indicating relatively stable simulations, while BC shows the lowest RMSE and most compact distribution, suggesting high accuracy. The NMB results show that the deviations of NO<sub>3</sub><sup>-</sup>, SO<sub>4</sub><sup>2-</sup>, and NH<sub>4</sub><sup>+</sup> are relatively constrained, although overestimations or underestimations still occur in some cities. In contrast, the NMB boxplots for OC and BC exhibit significant elongation, with deviations ranging from -50% to +150% in certain locations. The high bias of OC is mainly attributed to inaccuracies in the representation of SOA formation and precursor emissions, while the large relative deviations of BC are likely due to its low ambient concentrations, where even minor absolute errors can lead to amplified relative discrepancies.

A cross-model comparison reveals that CMAQ exhibits the highest correlation coefficients, particularly excelling in reproducing the spatial and temporal patterns of NO<sub>3</sub><sup>-</sup> and NH<sub>4</sub><sup>+</sup>. However, it shows a tendency to overestimate OC concentrations. CAMx shows intermediate performance across most components. While the EPICC-Model demonstrates relatively lower consistency in species simulation, yet maintains more stable performance in terms of RMSE and NMB.

### 3.3 AQI and pollution forecast accuracy

To comprehensively evaluate the applicability of the three CTMs in graded air quality forecasting, this study systematically assesses the performance of the EPICC-Model, CAMx, and CMAQ in the Air Quality Index (AQI) level prediction based on the *Technical guideline for numerical forecasting of ambient air quality* (HJ1130-2020). According to the evaluation criteria defined in the *Technical Regulation for the Assessment of Urban Ambient Air Quality Index (AQI) Forecasting (Trial)* (China National Environmental Monitoring Center, 2020), and considering the needs of operational forecasting, a prediction is deemed accurate if the observed AQI falls within ±25% of the forecasted value. It should be noted that, as dust events were not included in the simulations, the "heavy pollution" and "severe pollution" categories were combined in the subsequent analysis. Based on this criterion, all three models exhibit a consistent pattern across different pollutants and pollution levels: the higher the pollution level, the lower the forecasting accuracy. The algorithms and definitions of AQI and IAQI are provided in the Supplementary Material for reference.

In terms of overall forecasting accuracy, CAMx performes best in predicting AQI (84%), PM<sub>2.5</sub> (81%), and MDA8 O<sub>3</sub> (89%) levels, primarily attributed to its detailed representation of gas-phase precursor reaction chains under low concentration conditions. The EPICC-Model shows stable performance in forecasting good to moderate pollution levels, indicating strong capability in simulating regional pollution transformation and the early evolution stages of pollution processes. This is largely due to the EPICC-Model's incorporation of key heterogeneous reaction mechanisms that critically influence the rapid growth of secondary inorganic aerosols during heavy pollution episodes (EPICC-Model Working Group, 2025), including SO<sub>2</sub> manganese-catalyzed oxidation (Wang et al., 2021), N<sub>2</sub>O<sub>5</sub> heterogeneous hydrolysis (Yang et al.,

2024), and HONO heterogeneous formation (Zhang et al., 2022). CMAQ demonstrates superior performance in identifying moderate and higher pollution levels, particularly exhibiting notably higher accuracy than CAMx and the EPICC-Model for moderate to heavy and severe PM<sub>2.5</sub> levels. This advantage is attributed to its comprehensive aerosol-radiation-cloud feedback mechanisms, such as the AERO6 module, which provide detailed representation of pollutant accumulation and regional transport processes. However, all three models show generally low accuracy under extreme pollution scenarios, highlighting limitations in current chemical transport models for simulating nonlinear pollution buildup and the synergistic effects of extreme weather and pollution.

Table 5. Comparison of AQI level forecast accuracy for 337 Chinese cities.

| Model       | Excellent | Good | Light<br>Pollution | Moderate<br>Pollution | Heavy and<br>Severe Pollution | Accuracy |
|-------------|-----------|------|--------------------|-----------------------|-------------------------------|----------|
| EPICC-Model | 85%       | 85%  | 62%                | 39%                   | 23%                           | 82%      |
| CAMx        | 90%       | 87%  | 57%                | 31%                   | 11%                           | 84%      |
| CMAQ        | 83%       | 88%  | 63%                | 41%                   | 36%                           | 82%      |

**Table 6.** Comparison of PM<sub>2.5</sub> IAQI level forecast accuracy at 1,644 monitoring sites in China with observations.

| Model       | Excellent | Good | Light<br>Pollution | Moderate<br>Pollution | Heavy and<br>Severe Pollution | Accuracy |
|-------------|-----------|------|--------------------|-----------------------|-------------------------------|----------|
| EPICC-Model | 85%       | 69%  | 51%                | 35%                   | 25%                           | 79%      |
| CAMx        | 90%       | 68%  | 46%                | 26%                   | 11%                           | 81%      |
| CMAQ        | 83%       | 68%  | 50%                | 40%                   | 41%                           | 77%      |

**Table 7.** Comparison of MDA8 O<sub>3</sub> IAQI level forecast accuracy at 1,644 monitoring sites in China with observations.

| Model I     | Excellent | Good | Light     | Moderate  | Heavy and        | Accuracy |
|-------------|-----------|------|-----------|-----------|------------------|----------|
|             |           |      | Pollution | Pollution | Severe Pollution | Accuracy |
| EPICC-Model | 91%       | 84%  | 57%       | 23%       | 6%               | 87%      |
| CAMx        | 96%       | 84%  | 56%       | 23%       | 12%              | 89%      |
| CMAQ        | 85%       | 92%  | 60%       | 14%       | 6%               | 86%      |

Given the high sensitivity and risk posed by polluted weather and severe pollution events to public health and environmental management, this study further focuses on the forecasting performance of PM<sub>2.5</sub> and MDA8 O<sub>3</sub> pollution events. Based on metrics such as hit rate, false alarm rate, and the distance from the random operating characteristic (DROC) (definitions and calculation methods are provided in the Supplementary Material), the forecast effectiveness for general pollution conditions (PM<sub>2.5</sub> > 75  $\mu$ g m<sup>-3</sup> and MDA8 O<sub>3</sub> > 160  $\mu$ g m<sup>-3</sup>) and severe pollution conditions (PM<sub>2.5</sub> > 150  $\mu$ g m<sup>-3</sup>) is thoroughly analyzed to elucidate the applicability and potential limitations of each model under extreme pollution scenarios.

**Figure 8.** Comparison of CTMs performance based on hit rate and false alarm rate. Scatter points represent the performance of three models (EPICC-Model: circles, CAMx: diamonds, CMAQ: squares) under different pollution event thresholds:  $PM_{2.5} > 75~\mu g~m^{-3}$ ,  $PM_{2.5} > 150~\mu g~m^{-3}$ , and MDA8  $O_3 > 160~\mu g~m^{-3}$ . The red dashed line indicates the random forecast reference line. The DROC metric, which quantifies model skill, is provided in the lower-right corner, with higher values indicating better model performance.

Figure 8 visually compares the forecast performance of three models under different pollution thresholds using scatter points in ROC space. For general pollution scenarios ( $PM_{2.5} > 75 \mu g m^{-3}$ ), CMAQ shows the highest hit rate (64.7%) but is accompanied by a relatively high false alarm rate (9.6%), indicating a tendency toward systematic overestimation. CAMx achieves the lowest false alarm rate (4.9%), demonstrating stronger robustness, though its hit rate is also relatively low (45.0%), reflecting a conservative forecasting tendency. The EPICC-Model strikes a balance between hit rate (55.0%) and false alarm rate (7.6%), suggesting an optimized trade-off between accuracy and reliability. Under the more severe pollution condition ( $PM_{2.5} > 150 \mu g m^{-3}$ ), the hit rates of all models decrease significantly. While CMAQ still maintains the highest hit rate (26.8%), its miss rate reaches 73.2%. The EPICC-Model

and CAMx show hit rates of only 12.3% and 5.4%, respectively, indicating limited responsiveness of current models to extreme pollution events. Nevertheless, all models maintain false alarm rates below 2%. This reflects a conservative strategy that prioritizes avoiding false positives under extreme pollution conditions. For  $O_3$  pollution events (MDA8  $O_3 > 160 \,\mu g \,m^{-3}$ ), the EPICC-Model achieves the highest hit rate (45.6%), significantly outperforming CAMx (43.0%) and CMAQ (38.4%), while also maintaining a low false alarm rate (3.8%). This suggests advantages in its representation of  $O_3$  precursor transport, photochemical mechanisms, and boundary layer feedback processes. The DROC values provided in the lower-right table quantitatively reflect the above observations: CMAQ has the highest DROC (0.39) for general PM<sub>2.5</sub> pollution events, while the EPICC-Model performs best for  $O_3$  events (DROC=0.30). CAMx exhibits relatively low DROC across all thresholds, consistent with its conservative simulation style.

## 3.4 Capability in capturing regional persistent pollution events

To further evaluate the capability of the EPICC-Model, CAMx, and CMAQ in simulating trans-regional and long-duration pollution episodes, this study selects typical persistent PM<sub>2.5</sub> and O<sub>3</sub> events in 2021, and conducts a dedicated analysis incorporating pollutant transport pathways and the stability of meteorological fields.

**Figure 9.** Spatiotemporal distribution of the persistent PM<sub>2.5</sub> pollution event. Dots represent observed concentrations in 377 cities.

As shown in Figure 9, a persistent PM<sub>2.5</sub> pollution event occurred in eastern and central China from January 20 to 28, 2021. Initially, pollution was concentrated between 30° N and 40° N, forming a "dual-core" pattern over the North China Plain and the Sichuan Basin. By January 24, PM<sub>2.5</sub> concentrations peaked, with the pollution belt extending southeastward to the middle and lower Yangtze River regions and northeastward to the Songnen Plain. On January 28, under cold air influence, concentrations dropped below 75 μg m<sup>-3</sup> in most areas, leaving only isolated hotspots. All three models captured the temporal evolution from initiation to dissipation. The EPICC-Model showed superior performance in reproducing spatial gradients and temporal patterns, particularly in the centre of the North China Plain and Yangtze River Middle-Reach corridor. CAMx reproduced the northeastward transport path but underestimated pollution intensity, failing to capture the strong core in Henan during the peak. CMAQ systematically overestimated concentrations, likely due to chemical and physical parameterizations enhancing pollutant production and mix.

Figure 10. Time series of PM<sub>2.5</sub> concentrations in key cities along the pollution belt.

As shown in Figure 10, Zhengzhou recorded the first PM<sub>2.5</sub> peak (>150 μg m<sup>-3</sup>) on January 21, with the pollution plume spreading northeastward and causing a delayed concentration rise in Shenyang. All four cities (Zhengzhou, Shenyang, Shijiazhuang, and Beijing) reached their maxima, though peak timings varied. During the clearing phase, driven by southward cold air intrusion, concentrations declined across all cities, with faster removal in Zhengzhou and Shijiazhuang than in Shenyang and Beijing. All three models reproduced the overall process but showed systematic timing biases, with peaks in Shenyang and Zhengzhou simulated 24-48 hours earlier than observed. Such deviations may result from inaccuracies in the simulated wind field evolution within the meteorological models. In terms of magnitude, CAMx underestimated and CMAQ overestimated concentrations, while the EPICC-Model produced comparatively robust results.

Figure 11. Spatiotemporal distribution of the persistent O<sub>3</sub> pollution event.

As shown in Figure 11, a persistent  $O_3$  pollution episode occurred in China from June 3 to 14, 2021. The event began on June 3 with scattered pollution over the North China Plain and parts of Guangdong Province. By the peak stage on June 6, pollution intensity increased significantly, spreading to central, southeastern, and southern regions. By June 10, the pollution weakened and became more localized in North China and its western areas, while southern regions began clearing. By June 14, MDA8  $O_3$  concentrations dropped below  $100~\mu g$  m<sup>-3</sup> across most areas, leaving only a few localized hotspots in Northeast China and Inner Mongolia. Elevated temperatures, strong solar radiation, and stagnant atmospheric conditions favored  $O_3$  formation, whereas cold air intrusions promoted dispersion and removal.

During this persistent pollution event, all three models captured the full process of occurrence, development, and removal. Among them, the EPICC-Model best reproduced the spatial progression from central China to southeastern and southern regions and subsequent clearing toward the North China Plain and northeastern areas, although local overestimation occurred during the peak phase in the southeastern region, likely due to an excessively strong aerosol-radiation feedback. In contrast, CAMx and CMAQ systematically underestimated pollution intensity, particularly during the peak on June 6 and the initial clearing stage on June 10, failing to reproduce the strong pollution exceeding 160 µg m<sup>-3</sup> across Henan Province. This discrepancy may result from insufficient temporal resolution of emission inventories or limited photochemical sensitivity to precursors. Overall, the EPICC-Model outperformed the other two models in capturing spatiotemporal patterns, while CAMx and CMAQ showed deficiencies in reproducing peak concentrations.

Figure 12. Time series of O<sub>3</sub> concentrations in key cities along the pollution belt.

The pollution event occurred over central and eastern China between 30° N and 40° N. Qingdao first entered O<sub>3</sub> pollution on June 4, with MDA8 O<sub>3</sub> exceeding 160 μg m<sup>-3</sup>. The pollution then maintained high concentrations and expanded northward and southward, reaching Qingdao, Wuhan, Beijing, and Guangzhou by June 6. After June 9, the first clearing phase began from south to north, with Guangzhou improving first, followed by Wuhan and Qingdao, while the pollution center shifted to the North China Plain and northeastern regions, causing a sharp rise in Beijing around June 12. By June 14, most regions had cleared, with Beijing showing the fastest decline, likely due to enhanced cold air activity and higher boundary layer. All three models captured the full progression in Beijing and Qingdao accurately. For Wuhan and Guangzhou, where peak concentrations persisted, model predictions showed approximately one-day deviations, likely due to limited responses to abrupt meteorological changes.

## **4 Conclusions**

 This study presents a systematic evaluation of three CTMs (EPICC-Model, CAMx, and CMAQ) for PM $_{2.5}$  and MDA8 O $_3$  simulations over China in 2021, using unified WRF meteorological fields and multi-source emission inventories. The work fills a critical research gap by providing the first comprehensive comparison of the EPICC-Model against established CTMs. The results indicate that all three models can effectively reproduce the spatiotemporal evolution characteristics of PM $_{2.5}$  and MDA8 O $_3$  (PM $_{2.5}$ : R = 0.79-0.85, MDA8 O $_3$ : R = 0.91-0.94). In PM $_{2.5}$  simulations, the EPICC-Model shows seasonal turning points, with spring and summer results tending toward the underestimation trend of CAMx, while autumn and winter results align more closely with the overestimation tendency of CMAQ. Spatially, it exhibits higher consistency (annual IOA = 0.80), particularly in winter and over heavily polluted North China, effectively capturing pollution patterns and peaks. For O $_3$ , the EPICC-Model performs

734

736737

738

739740

747

749

758

770

772

well during high-concentration summer episodes, accurately reproducing extremes, thanks to enhanced heterogeneous HONO formation and nitrate photolysis, which increase OH radicals and accelerate VOC oxidation, and the CB6r5 chemical mechanism that better represents biogenic VOC oxidation. Furthermore, the EPICC-Model produces particularly accurate O3 simulations over the eastern, central, and Chengdu-Chongqing regions. Notably, all three models share common deficiencies in PM<sub>2.5</sub> simulations. Systematic underestimations occur in arid northwestern China due to unaccounted dust processes, highlighting the critical influence of natural emissions on model accuracy. Conversely, autumn-winter overestimations prevail in North China and the Sichuan Basin, attributable to the ACM2/YSU boundary layer schemes underestimating mixing layer height and weakening nocturnal vertical diffusion, thereby inadequately reproducing inversion layers and pollution accumulation. Uncertainties in emission inventories further exacerbate regional biases, such as potential overestimation of anthropogenic emissions in the Sichuan Basin and underestimation of agricultural NH<sub>3</sub> emissions in Henan, which affects nitrate aerosol formation. Future improvements should incorporate dust processes, refine boundary layer parameterizations to better simulate pollutant accumulation under stable meteorological conditions, and employ higher-resolution emission inventories with improved temporal variability characterization.

In terms of PM<sub>2.5</sub> component simulations, the EPICC-Model, CAMx, and CMAQ all reasonably reproduce the chemical composition characterized by NO<sub>3</sub><sup>-</sup> dominance and associated regional variations. Compared with CAMx and CMAO, the EPICC-Model exhibits smaller biases in SO<sub>4</sub><sup>2-</sup> and NH<sub>4</sub><sup>+</sup>, with more robust control of absolute concentrations as reflected in RMSE and NMB, indicating a relatively consistent overall performance, although its correlation metrics are slightly lower than the other two models. Nevertheless, all three models share common limitations: SO<sub>4</sub><sup>2-</sup> is generally underestimated, leading to insufficient formation of (NH<sub>4</sub>)<sub>2</sub>SO<sub>4</sub> and lower NH<sub>4</sub><sup>+</sup> concentrations; SOA formation remains highly uncertain, influenced by precursor VOC emissions, oxidation pathways, and gas-particle partitioning parameterizations, resulting in spatial deviations in OC; BC simulations are highly sensitive to emission inventories, occasionally leading to slight overestimations in certain cities. Addressing these issues in future work should focus on optimizing secondary inorganic aerosol chemistry, improving SOA formation mechanisms and precursor emissions, and enhancing the resolution and accuracy of emission inventories to improve the models' capability in reproducing PM<sub>2.5</sub> chemical composition.

Regarding AQI level classification ability, CAMx achieved the highest overall accuracy (84%), demonstrating particular strengths in distinguishing "excellent" and "good" air quality levels. CMAQ is relatively accurate in pollution level identification, with the highest hit rate (64.7%) in identifying general pollution where  $PM_{2.5} > 75~\mu g$  m<sup>-3</sup>, but it has a high false alarm rate (9.6%), indicating a tendency toward overforecasting. The EPICC-Model performs exceptionally well in identifying  $PM_{2.5}$  light to moderate pollution levels and MDA8 O<sub>3</sub> general pollution, with a hit rate of 45.6% for MDA8 O<sub>3</sub> > 160  $\mu g$  m<sup>-3</sup>, higher than CAMx (43.0%) and CMAQ (38.4%). Furthermore, the forecast accuracy of all three models for single extreme pollution

events was less than 2%, indicating that the response capability of existing models to sudden pollution events still needs further improvement. Notably, the EPICC-Model demonstrated relatively balanced false alarm rates across all pollution types, indicating its capability to simulate pollution processes comparable to mainstream models, particularly showing strong potential in responding to light to moderate pollution and O<sub>3</sub> pollution.

In the simulation analysis of typical persistent PM<sub>2.5</sub> and O<sub>3</sub> pollution events in 2021, the EPICC-Model demonstrated strong capabilities, particularly outperforming CAMx and CMAQ in reproducing spatial distributions and pollution evolution. The EPICC-Model successfully captured the entire process of pollution occurrence, development, and clearance, particularly in the core area of the North China Plain and the diffusion zone of the middle Yangtze River, where the simulation results were in close agreement with observational data. CAMx and CMAQ, however, performed less effectively in capturing persistent pollution events, both exhibiting significant deviations during the simulation of severe pollution phases. CAMx underestimated pollution intensity during severe pollution events, while CMAQ responded too rapidly during the dissipation phase, leading to premature pollution decay (the average duration of the pollution process was underestimated by 12-18 hours).

This study establishes a multi-model evaluation framework to systematically compare the applicability and robustness of the EPICC-Model against internationally established models over China. By identifying both model-specific limitations and common challenges shared across different models, this work provides a clear pathway for improving the next generation of atmospheric CTMs. The results not only offer critical scientific support for air pollution control and policy-making in China, but also serve as a valuable reference for other developing countries facing similar environmental challenges. This study contributes to promoting the collaborative development of global air quality modeling techniques and advancing sustainable environmental governance.

## Code and data availability

The models utilized in this study are open source and publicly available. The EPICC-Model code can be found at <a href="https://earthlab.iap.ac.cn/resdown/info\_388.html">https://earthlab.iap.ac.cn/resdown/info\_388.html</a>, the CMAQ model code is available at <a href="https://github.com/USEPA/CMAQ/tree/5.3.3">https://github.com/USEPA/CMAQ/tree/5.3.3</a>, the CAMx model code can be accessed at <a href="https://github.com/NCAR/WRFV3/releases/tag/V3.9">https://github.com/NCAR/WRFV3/releases/tag/V3.9</a>. All air quality observations used in this study were obtained from the China National Environmental Monitoring Centre. In addition, the sources of the emission inventories and meteorological data used for model evaluation are clearly described in the main text and Supplementary Information. The remaining primary datasets of this study have been deposited in the ZENODO repository (Lou and Wu, 2025). Additional relevant data are available from the corresponding author upon reasonable request.

#### **Author contributions**

ML, QW, WengdW, and HC conceptualized and organized the study. ML

- performed the data analysis and drafted the manuscript. ML and XF conducted the
- numerical simulations, and FY optimized the simulation code. XF and WeiW provided
- data support. DL contributed to the meteorological model evaluation. ML, QW,
- WengdW, and HC validated and analyzed the simulation results. KC, JZ, and ZW
- reviewed the research findings and provided key feedback. All authors read and
- approved the final manuscript.

### Competing interests

827

The authors declare no competing interests.

### Acknowledgements

- We appreciate the technical support provided by the National Large Scientific and
- Technological Infrastructure, the "Earth System Numerical Simulation Facility"
- (https://cstr.cn/31134.02.EL).

### Financial support

- This study was supported by the National Key Research and Development Program
- of China (Grant No. 2023YFC3705705 and 2024YFC3713603) and the Strategic
- Priority Research Program of the Chinese Academy of Sciences (Grant No.
- XDB0760400).

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
