# Peer review of "Evaluating the EPICC-Model for Regional Air Quality"

_EGUsphere, 2025_

## Referee Comment (RC1)

**Review of egusphere-2025-4441**

**Evaluating the EPICC-Model for Regional Air Quality Simulation: A Comparative Study with CAMx and CMAQ**

General Comments

This paper intercompares ozone and PM2.5 simulation results over China for the year 2021 between a new photochemical grid model developed for China (EPICC-Model) and two established models (CAMx and CMAQ). The analyses go deep into the performance comparisons spatially and temporally across many scales, yielding a broad assessment of the EPICC-Model relative to the other two models on a regional basis. Based on how the modeling study was conducted, the conclusions are generally sound. The paper is well-written with proper English grammar and syntax, with just a few exceptions as noted below.

Overall, model performance in replicating spatiotemporal variations of monitoring data appears to be more similar than different among the 3 models, which suggests that errors/uncertainties in input data may dominate over differences in model formulation. An additional analysis would be helpful to quantitatively compare inter-model differences relative to model-observation differences. The configurations of CMAQ and CAMx should be made more consistent with the EPICC-Model to minimize controllable factors that obscure the reasons for inter-model differences in ozone and PM2.5 results. Nevertheless, some substantial model differences do occur in a few cities, which are briefly reviewed with some potential reasons offered simply as conjecture. More in-depth isolated analyses for specific urban centers would be useful, perhaps using process analysis techniques, to quantitatively understand the specific causes for those differences.

At the time of my review, the code and data availability link at https://earthlab.iap.ac.cn/resdown/info_388.html does not appear to allow access to the source code, at least not easily. It would be helpful if the authors put EPICC-Model source code on Zenodo.

Based on these limitations and the key issues I note below, major revisions are needed before publication.

Specific Comments

1.  A) The paper stresses systematic consistency in the model comparison. But according to Table 1 that is not entirely accurate. The EPICC-Model uses 20 vertical layers, while CAMx and CMAQ use 14. The authors stress the better layer structure in EPICC-Model as "demonstrating superior vertical resolution compared to the 14 layers coordinates used in other two models" especially "for near-surface turbulent mixing and nocturnal stable layer structures." CAMx and CMAQ should use the same 20 layer structure as EPICC-Model to improve model-to-model consistency and remove a key source of uncertainty in the comparison of model results. If a single WRF run was used to drive all 3 models (run with 30 layers as shown in Table 2), use of consistent layer structures is possible. While CAMx allows for sub-setting WRF layers to fewer CAMx layers, the CMAQ MCIP preprocessor does not allow for this and requires the use of the full 30 layer WRF structure. So it is not clear how the authors transmitted 30-layer WRF meteorological fields to a subset of 14 CMAQ layers, unless they ran a separate configuration of WRF with 14 layers or developed their own WRF-to-CMAQ meteorological interface. Additionally, the authors state that both CMAQ and CAMx use "default" boundary conditions to specify concentrations on the 45-km outer grid. These values and their sources should be stated in the paper with a rationale justifying their selection. Both CMAQ and CAMx can utilize the same MOZART global model output data to derive boundary conditions, as was done for EPICC-Model. While I understand that boundary conditions do not appear to have appreciable effects on model results overall, use of consistent boundary conditions is preferable.

B) A comparison of EPICC-Model and CAMx in Table 1, and the descriptions of these models in the text, indicate that most EPIC-Model algorithms are similar or identical to CAMx algorithms. The main difference is in the chemical treatments. CAMx was run with the CB05 photochemistry mechanism, which is an odd choice given that it represents 20-year old science. CAMx v7.00 used in this study includes CB6r4, and CAMx v7.10 released just 6 months later in 2020 includes CB6r5 (the same CAMx mechanism that was adopted by EPICC-Model). This is a critical issue and CAMx should be rerun with CB6 chemistry. If the MEIC VOC emissions were uniformly speciated to CB6 compounds for all 3 models, then that speciation would lead to missing VOC mass in CB05 due to newer CB6 species that are unrecognized by CB05. That alone would lead to different photochemical responses. A full explanation is needed on how MEIC emissions were temporally, spatially, and chemically processed for each of the 3 models.

C) Additionally, several of the simple descriptions for CAMx processes in Table 1 are inadequate or inaccurate. The PPM vertical advection solver was not yet available in CAMx v7.00 or v7.10, so that entry in Table 1 is either incorrect, or the authors used CAMx v7.20 or later. It is inaccurate to state that CAMx employs the YSU vertical diffusion scheme, rather it employs a first-order K-theory approach (apparently like EPICC-Model) where the vertical diffusivities are diagnostically derived from WRF meteorological fields based on the YSU approach. CAMx v7.00 does not use the original SOAP scheme of Strader et al. (1999), but an updated SOAP2 scheme unique to CAMx (refer to the CAMx v7 user's guide). CAMx uses the full-science TUV as a pre-processor to generate a photolysis rate input file, then internally applies a streamlined TUV version to adjust those for effects from clouds and aerosols. This identical TUV process was adopted in EPICC-Model (according to Wang et al., 2025: Development and evaluation of photolysis and gas-phase reaction scheme in EPICC-model: Impacts on tropospheric ozone simulation, *Atmos. Environ.*, https://doi.org/10.1016/j.atmosenv.2025.121373).

2. Abstract, Line 41: I didn't see specific references or comparisons of EPICC-Model performance "against international benchmarks" in the body of the paper. Do you mean "against international models"?

3. Line 56: I suggest adding a phrase like "due to a non-linear response to precursor NOx reductions" or similar to add clarity about the cause of increasing urban ozone.

4. Numerous locations: Certain words are used throughout the paper that carry vague meaning, most frequently the word "stable". For example, Lines 199-200, "Simulations of 10 m wind speed were generally stable", and again on Line 212, "Surface pressure simulations showed strong stability." It is unclear what the word stable is meant to convey. I have not seen other uses of "stable" in the context of model performance evaluation, so I cannot suggest an alternative word. Other occurrences:

   Lines 338: "EPICC-Model … reflecting a more balanced and stable performance overall."

   388: "CAMx is stable (IOA=0.78) and suitable for multi-seasonal averages."

   426: "Comprehensive evaluation indicates that the EPICC-Model shows stable seasonal MDA8 O3 performance"

   437: "Correlation coefficients exceed 0.75 for all models, while RMSE and STD remain relatively low, indicating higher stability and accuracy in MDA8 O3 simulations compared to PM2.5."

540: "BC exhibits the most compact R distribution, with low inter-city variability, reflecting stable and consistent simulation by all models."

545: "$SO_4^{2-}$ and $NH_4^+$ exhibit moderate RMSE, indicating relatively stable simulations"

579: "The EPICC-Model shows stable performance in forecasting good to moderate pollution levels"

645: "incorporating pollutant transport pathways and the stability of meteorological fields."

5.  Line 214: A surface pressure MB of 20 hPa (20 mb!) is extraordinarily large and highly concerning. Perhaps this is a typographical error and the authors meant 2 hPa (2 mb)?

6.  Lines 229-235: The authors appropriately argue about the applicability of different emission inventories representing different years. On Line 230, "emission inventories do not exactly match the simulation year", I suggest the removal of "exactly" as it's clear they do not match. I agree with this statement: "Under the assumption of no substantial changes in regional climate and socioeconomic activities, such inventories can reasonably represent the emission patterns of the study period." However, I suggest adding an explicit statement to the effect of "No projections of the individual emission inventory data to the 2021 simulation year were performed" to add clarity that the emission inventories were used without modification or alignment.

7.  Lines 291-292: "In contrast, the EPICC-Model shows greater adaptability…" This statement is not clear, although I think I understand the intent. Perhaps a better word like "variability" or "spatial/temporal response" would be better than "adaptability" to reflect the authors' meaning.

8.  Lines 313-317: "This discrepancy mainly stems from differences in the treatment of photolysis and chemical mechanisms among the models" and "In addition, the EPICC-Model employed the CB6r5 chemical mechanism, which offered more comprehensive representation of BVOC oxidation (especially isoprene) compared to CB6r3 (used in CMAQ) and CB05 (used in CAMx), thereby increasing O3 formation potential." Certainly true – see my comments above about running CAMx with CB05 rather than CB6r5. Yet the photolysis treatments in EPICC-Model and CAMx are apparently identical so the reference to different photolysis should be removed.

9.  Lines 403-405: "CAMx generally underestimated O3 nationwide, likely due to limitations of the CB05 mechanism and its coarse $14\sigma p$ vertical resolution." See my comments above. The selection of CB05 and 14 vertical layers are the authors' choice, not fixed features of CAMx or CMAQ.

10. Lines 411-412: "Given the use of identical meteorological fields". This is untrue, see my comments about Table 1 above. Meteorology is not identical using different layer structures and boundary layer treatments.

11. Figure 5 and associated text: Please include a description of Taylor diagrams and how to interpret them. What does standard deviation (STD) refer to, the modeled variability or model-measurement variability, and what are the units? Why is STD=1 the best? The diagrams appear to be missing value labels for RMSE.

12. Line 446: "CMAQ demonstrating greater sensitivity to pollution peaks". The word sensitivity seems inappropriate here, perhaps better to say "greater ability to replicate pollution peaks". Also, Line 448 "indicating enhanced error sensitivity", perhaps simply remove the word "sensitivity" because including it makes the meaning unclear.

13. Lines 478-481: "The insufficient formation of $SO_4^{2-}$ not only limits the production of ammonium sulphate [(NH4)2SO4] but also reduces the consumption potential of NH3 (Gao et al., 2018), thereby suppressing the simulated concentration of ammonium (NH4+)." Certainly true. This also frees up more NH3 to bond with HNO3, driving up nitrate levels – one possible reason for nitrate over predictions.

14. Lines 576-579: "CAMx performes best in predicting AQI (84%), PM2.5 (81%), and MDA8 O3 (89%) levels, primarily attributed to its detailed representation of gas-phase precursor reaction chains under low concentration conditions." The word "performes" is misspelled. Attempting to attribute CAMx performance in this manner is conjecture and unclear, especially given the use of CB05. I suggest removing such conjecture if concrete reasoning backed by appropriate evidence (such as process analysis) and/or a reference cannot be provided.

15. Lines 580-581: "indicating strong capability in simulating regional pollution transformation and the early evolution stages of pollution processes." What is meant by "transformation" and "early evolution stages" here? Please explain or simply revise this sentence to be more specific.

16. Lines 589-592: "This advantage is attributed to its comprehensive aerosol-radiation-cloud feedback mechanisms, such as the AERO6 module, which provide detailed representation of pollutant accumulation and regional transport processes." CMAQ AERO6 is stated, but Table 1 indicates that CMAQ was run with AERO7. Like my comment on Lines 576-579, I suggest removing such conjecture about the attribution of CMAQ performance without concrete evidence (process analysis results).

17. Lines 621-622: "CAMx achieves the lowest false alarm rate (4.9%), demonstrating stronger robustness." The word robustness seems to be inappropriate in this context (like "stable" examples above). More simply, the false alarm rate is low because CAMx underpredicts PM2.5 levels and does not predict high PM events.

18. Lines 623-625: "The EPICC-Model strikes a balance between hit rate (55.0%) and false alarm rate (7.6%), suggesting an optimized trade-off between accuracy and reliability." The trade-off is true, but it cannot be considered purposely "optimized". I suggest removing the word "optimized" as it is misleading.

19. Lines 630-631: "This reflects a conservative strategy that prioritizes avoiding false positives under extreme pollution conditions." This statement is probably lost in translation. The models do not present a strategy of any kind, especially that avoids certain performance issues – they are objective methods based on the science algorithms they employ. Perhaps a more accurate statement would be, "This reflects an inability of these models to simulate extreme pollution conditions".

20. Lines 632-633: "the EPICC-Model achieves the highest hit rate (45.6%), significantly outperforming CAMx (43.0%) and CMAQ (38.4%)." Be careful using words like "significantly" in a qualitative and subjective sense, as readers will have different ideas on what "significant" means to them. I suggest removing that word, especially since a difference of 2-7% seems small and likely not statistically significant (unless you report a statistical hypothesis test with a set p-value).

21. Lines 634-636: "This suggests advantages in its representation of O3 precursor transport, photochemical mechanisms, and boundary layer feedback processes." Again, like my comment on Lines 576-579, I suggest removing such conjecture about the attribution of EPICC-Model performance without concrete evidence (process analysis results), especially given small differences

in the hit rate statistics and many similarities in model treatments (photochemical mechanisms, boundary layer dynamics) and source of inputs.

22. Lines 639-640: "CAMx exhibits relatively low DROC across all thresholds, consistent with its conservative simulation style." Like my comments for Lines 630-631, CAMx is not developed to possess a "style" of any kind. I suggest removing the last part of the sentence starting at "consistent".

23. Lines 657-658: "The EPICC-Model showed superior performance in reproducing spatial gradients and temporal patterns." The word "superior" is too strong. Given results in Figure 9 it's difficult to assess this visually. I suggest replacing "superior" with "better" or "improved".

24. Lines 661-662: "likely due to chemical and physical parameterizations enhancing pollutant production and mix". I suggest removing this conjecture for similar reasons provided above.

25. Lines 674-676: "In terms of magnitude, CAMx underestimated and CMAQ overestimated concentrations, while the EPICC-Model produced comparatively robust results." Except for Zhengzhou, all 3 models look to track very similarly. Perhaps make the quoted statement specific to Zhengzhou and then say all models otherwise performed similarly.

26. Lines 694-696: "In contrast, CAMx and CMAQ systematically underestimated pollution intensity, particularly during the peak on June 6 and the initial clearing stage on June 10." From Figure 11, the contoured predictions match the color of the measured dots better in lower ozone areas on June 6 and 10, especially in central China. It looks to me that EMICC-Model tends to overpredict ozone in those areas.

27. Lines 697-699: "This discrepancy may result from insufficient temporal resolution of emission inventories or limited photochemical sensitivity to precursors." Is this possible? I thought all 3 models were given the same emissions. Photochemical sensitivity should not be substantially different between the various CB chemistry mechanisms among the models. Again, I suggest removing such conjectures without solid evidence from process analyses. Figure 12 does not show much difference between the models except in Guangzhou where CMAQ seemed to perform best and the other 2 models under predicted substantially.

28. Lines 769-770: "The EPICC-Model performs exceptionally well in identifying PM2.5 light to moderate pollution levels and MDA8 O3 general pollution." I suggest removing the subjective word "exceptionally" without a numerical example to provide evidence for it. See also my comments above for Lines 632-633 with reference to the small 2-7% differences in hit rate.

Technical Corrections

Be sure to define all acronyms at the time of their first occurrence. There are many undefined acronyms throughout the text.

Line 25: Remove the unneeded word "the" in the phrase "the EPICC-Model".

Line 26 (and elsewhere): What is meant by "unified" in this context? Either use a more appropriate adjective or remove entirely.

Line 28: The meaning of "high spatial consistency" is not clear in this limited context, perhaps simply replace with "high spatial consistency with measurements" if that is what is meant.

Line 34: Perhaps a better word for "identifying" is "forecasting".

Line 36: Suggest removing "compound" because it is unclear without additional context.

Line 65: Replace "ENVIRON Corporation" with "Ramboll" as the former company no longer exists.

Line 129: Replace "(Anon, 2020)" with "(Ramboll, 2020)" and update in the reference list.

Line 132: Replace "(Anon, 2021)" with "(EPA, 2021)" and update in the reference list.

Line 348: Replace "错误!未找到引用源" with English. I think it is a reference to Table 3.

Line 558-560: "While the EPICC-Model demonstrates relatively lower consistency in species simulation, yet maintains more stable performance in terms of RMSE and NMB." I suggest removing either the word "While" or "yet" to make the syntax correct.